# Atypical Gait Cycles in Parkinson’s Disease

**DOI:** 10.3390/s21155079

**Published:** 2021-07-27

**Authors:** Marco Ghislieri, Valentina Agostini, Laura Rizzi, Marco Knaflitz, Michele Lanotte

**Affiliations:** 1Department of Electronics and Telecommunications, Politecnico di Torino, 10129 Turin, Italy; valentina.agostini@polito.it (V.A.); marco.knaflitz@polito.it (M.K.); 2PoliToBIOMed Lab, Politecnico di Torino, 10129 Turin, Italy; 3Department of Neuroscience “Rita Levi Montalcini”, University of Turin, 10126 Turin, Italy; l.rizzi@unito.it (L.R.); michele.lanotte@unito.it (M.L.); 4AOU Città della Salute e della Scienza di Torino, 10126 Turin, Italy

**Keywords:** foot-floor contact, human locomotion, Parkinson’s disease, statistical gait analysis, UPDRS

## Abstract

It is important to find objective biomarkers for evaluating gait in Parkinson’s Disease (PD), especially related to the foot and lower leg segments. Foot-switch signals, analyzed through Statistical Gait Analysis (SGA), allow the foot-floor contact sequence to be characterized during a walking session lasting five-minutes, which includes turnings. Gait parameters were compared between 20 PD patients and 20 age-matched controls. PDs showed similar straight-line speed, cadence, and double-support compared to controls, as well as typical gait-phase durations, except for a small decrease in the flat-foot contact duration (−4% of the gait cycle, *p* = 0.04). However, they showed a significant increase in atypical gait cycles (+42%, *p* = 0.006), during both walking straight and turning. A forefoot strike, instead of a “normal” heel strike, characterized the large majority of PD’s atypical cycles, whose total percentage was 25.4% on the most-affected and 15.5% on the least-affected side. Moreover, we found a strong correlation between the atypical cycles and the motor clinical score UPDRS-III (*r* = 0.91, *p* = 0.002), in the subset of PD patients showing an abnormal number of atypical cycles, while we found a moderate correlation (*r* = 0.60, *p* = 0.005), considering the whole PD population. Atypical cycles have proved to be a valid biomarker to quantify subtle gait dysfunctions in PD patients.

## 1. Introduction

Parkinson’s disease (PD) is a neurodegenerative condition that can cause multiple impairments, notably in motor function, due to rigidity, bradykinesia, tremor at rest, and postural instability with frequent falling [1,2,3]. Although gait impairments are among the most common and disabling symptoms of PD patients, gait is not routinely assessed quantitatively, but it is described in general terms [4]. Indeed, to diagnose PD in a clinical setting, clinicians generally consider clinical manifestations, such as motor and non-motor symptoms, and rate disease severity based on the Unified Parkinson’s Disease Rating Scale (UPDRS) [5]. In particular, they evaluate the motor performance of PD patients through clinical motor examination, using the UPDRS-III score. Such clinical assessment largely depends on the expertise of the clinicians, and it is subjective, leading to variations in assessment between clinicians [6].

Instrumented gait analysis has revealed great potential in the objective assessment of PD locomotor performance [7,8]. In recent years, there has been increased interest in exploring new sensors and methods to obtain motor outcome measures in PD populations [9,10,11]. However, both in the diagnosis and in the management of PD patients, it is of the uttermost importance to find appropriate and reliable motion biomarkers [9], especially in the first stages of the pathology [12,13]. In particular, there is evidence that the foot and lower leg segments better classify motor anomalies than any other segment [14]. Indeed, in the last decade, there has been a sustained research activity in the study of foot strike variability [15], foot clearance [12,16,17] (i.e., the height of the foot above ground during the swing phase), heel-to-toe motion characteristics [18], as well as stride-to-stride variability [16], in cohorts of PD patients.

From a methodological point of view, walking mats [19] and different foot-worn or shoe-type sensors [20] have been used, including foot-pressure insoles [18], foot-switches [21], and Inertial Measurement Units (IMUs) [22]. The use of a continuous walking protocol instead of short intermittent walks, with no fewer than 30 steps (≥50 steps optimal), was suggested to obtain reliable results when calculating gait variability measurements [19]. Some walking patterns, such as turning have been found to be affected, even in the early stages of PD, with increased turning arcs, time to complete the turn, and a larger number of steps taken to complete the turn [23,24]. It has been suggested that turning/curved walking is more likely to cause gait instabilities and increased variability compared to straight walking [23,24,25]. In particular, it was found that high-functioning and well-treated PD patients, exhibiting no changes in speed during straight-line walking, increased their gait variability during curved walking [26]. Although acceleration/deceleration and turning strides are often ignored in the large majority of works analyzing PD gait, some researchers introduced in their protocols trajectories, including both straight-line and curved trajectories [21] (e.g., 4 × 10 m walk [27,28]).

Previous works using Statistical Gait Analysis (SGA) also rely on sessions of continuous walk, recording foot-switch signals for several minutes [29,30,31], along a path including both straight-line walking and turnings. However, the analysis is then usually limited to straight-line walking alone. An important procedure at the base of this technique is the automatic segmentation and classification of gait cycles, based on the foot-floor contact sequence adopted by a subject during locomotion [21]. Gait cycles may belong to distinct “classes”, characterized by a different sequence of gait phases. In healthy individuals, the most common gait cycle is HFPS, defined as the physiological sequence of Heel Contact (H), Flat Foot Contact (F), Push Off (P), and Swing (H), with H, F, and P being the sub-phases of stance [30]. HFPS is considered the typical gait cycle. By contrast, it is atypical any other gait cycle showing a foot-floor contact sequence different from HFPS. As an example, a higher percentage of atypical gait cycles was reported in cerebral palsy (CP) children, compared to normally developing children: The effect was more evident on the hemiplegic side, but it was also documented on the contralateral side [32]. The most frequently observed atypical cycles were PFPS and PS cycles, in which the forefoot, instead of the heel, first touched the ground at initial contact. An abnormal percentage of atypical cycles was also documented in patients who underwent total hip arthroplasty (THA), one year after surgery [33]. Both the above-mentioned studies [32,33], suggested the potential of using atypical cycles as biomarkers in detecting subtle gait abnormalities. Nevertheless, these works did not present any in-depth study of foot-floor contact sequence and atypical cycles, since they were mainly focused on electromyographic (EMG) activity patterns of muscles.

To the best of our knowledge, there are no studies focusing on the characterization of atypical gait cycles in PD patients. Furthermore, there is a lack of studies that have considered a continuous and prolonged overground walk lasting several minutes, including in the analysis of both straight gait and turnings. On the one hand, this kind of analysis seems promising for obtaining objective and reliable gait biomarkers for assessing motor fluency in natural walking conditions. On the other hand, UDPRS-III is currently the score most used by clinicians to evaluate the motor performance of PD patients. Therefore, it is important to validate new objective, measurement-based, biomarkers against this clinical score.

This study aims at describing the differences in foot-floor contact sequences between PD patients and healthy individuals, analyzing the typical and atypical gait cycles extracted from the foot-switch signals acquired during a five-minutes continuous walk. Moreover, the clinical validity of the percentage of atypical gait cycles in the assessment of gait in Parkinson’s Disease was checked through a correlation analysis between the percentage of atypical gait cycles and the motor score UPDRS-III.

## 2. Materials and Methods

### 2.1. Sample Population and Experimental Protocol

This study involved 20 patients affected by PD (age: 59.1 ± 8.4 years; height: 175.9 ± 9.5 cm; weight: 77.6 ± 12.3 kg) and 20 healthy volunteers (age: 55.2 ± 9.6 years; height: 168.1 ± 9.5 cm; weight: 70.9 ± 15.7 kg). PD patients were selected among those eligible for Deep Brain Stimulation (DBS) at the Stereotactic and Functional Neurosurgery Unit of the University of Turin (Turin, Italy), between October 2018 and October 2020.

The inclusion criteria were:the diagnosis of PD, according to the UK Brain Bank principles;a good response to levodopa;medication-resistant motor fluctuation and dyskinesia;age at surgery under 70 years;absence of freezing of gait and postural instability unresponsive to pharmacological therapy;absence of dementia or severe cognitive impairment, psychiatric or behavioral disturbances;absence of abnormalities at cerebral MRI or relevant condition that increase surgical risk;the ability to walk independently for a few minutes without walking aids or external support, within the pharmacological best-ON time window.

The exclusion criterion was the presence of co-morbidities potentially affecting gait performance. PD patients’ features were: a disease duration of 12.8 ± 3.7 years, an average UPDRS-III score of 17.4 ± 8.1 (best-ON condition), and an average Hoehn Yahr score of 1.7 ± 0.4 (best-ON condition), respectively. Control subjects were enrolled among the patients’ caregivers, excluding those reporting neurological or musculoskeletal disorders affecting gait. Neuropsychological and behavioral assessments were performed for both control subjects and PD patients (during their best-ON condition). The Mini-Mental State Examination (MMSE) [34] was assessed as a screening measure of global cognitive functioning. More specifically, a test battery assessed five cognitive domains:reasoning through Raven Color Matrices (CPM47) test [35];memory through Bisyllabic Words Repetition (BWR), Corsi’s Block Tapping (CBT), and Rey auditory verbal learning (AVLT) tests [36,37];attention through Digit Cancellation Test (DCT) and Trail Making A (TMA) test [37,38];frontal executive functions through Trail Making B (TMB), Frontal Assessment Battery (FAB), and Clock Drawing Test (CDT) [39,40];phonemic and category verbal fluency [35].

According to the MMSE, all the control subjects and PD patients involved in the study were classified as normal in terms of global cognitive functioning (i.e., MMSE score higher than 25).

All the subjects were asked to walk barefoot for approximately 5 min at self-selected speed, back and forth over a 9-m straight walkway. PD patients were acquired during their best-ON conditions. Figure 1 shows a schematic representation of the walking path.

The experimenter manually timed each subject’s passage during straight walking (within the red marks). The average walking speed was defined as the total distance walked along the straight path divided by the total time required to go through it [41].

This study was approved by the Ethics Committee of A.O.U. Città della Salute e della Scienza di Torino—A.O. Ordine Mauriziano—A.S.L. “Città di Torino” (No. 0092029 approved on 11 September 2018). Written informed consent was obtained from each participant before the experimental sessions, and all the acquisitions were performed in accordance with the Declaration of Helsinki.

### 2.2. Data Acquisitions

Subjects were equipped, bilaterally, with foot-switches (size: 10 mm × 10 mm × 0.5 mm; activation force: 3 N), fixed under each barefoot sole through double-sided adhesive tape. Foot-switches were placed beneath the heel, the first, and the fifth metatarsal heads of each foot [21]. “Basographic” signals were recorded through the STEP32 multichannel acquisition system (Medical Technology, Turin, Italy) to detect the foot-floor contact events and determine the gait phases.

Figure 2A shows an example of foot-switch placement for a representative healthy subject. First, the foot-switches were placed under each bare foot by means of double-sided adhesive tape. Then, each foot-switch sensor was connected to the multichannel acquisition system through a connector fixed to the subject’s leg by means of an elastic belt. Figure 2B, represents an example of a foot-switch signal acquired from the left foot (more-affected side) of a representative PD patient of the sample population with the indication of the four gait phases (H: Heel Contact, F: Flat-Foot Contact, P: Push-Off, and S: Swing).

The sampling frequency was 2 kHz. The signals were then imported into MATLAB^®^ release 2020b (The MathWorks Inc., Natick, MA, USA) to be processed through custom routines.

### 2.3. Gait Phases Analysis

The pioneering work by Jackline Perry firsly introduce the Rancho Los Amigos (RLA) nomenclature, defining the gait phases in normal and pathological function as initial contact and load acceptance, mid and terminal stance, pre-swing, initial swing, mid swing, and terminal swing [42]. In this work, we specifically focus on foot-floor contact gait phases, as directly measurable through foot-switches.

As a first step, foot-switch signals were filtered through an anti-causal anti-bounce filter to remove spurious spikes due to switch bounces, and to obtain a 4-level foot-switch signal modeling the following four gait phases [21]:i.Heel contact (H): only the foot-switch under the heel is closed;ii.Flat-foot contact (F): the heel foot-switch and one or both the metatarsal foot-switches are closed;iii.Push-off (P): one or both the metatarsal foot-switches are closed;iv.Swing (S): none of the foot-switches is closed.

Filtered foot-switch signals were then automatically segmented into gait cycles and classified in different categories, based on the gait-phase sequence [33]. More specifically, gait cycles showing the physiological sequence of Heel Contact (H), Flat Foot Contact (F), Push Off (P), and Swing (H) phases were classified as typical gait cycles. By contrast, all gait cycles showing a foot-floor contact sequence different from HFPS were classified as atypical.

Figure 3A shows an example of the 4-level foot-switch signal acquired during a typical gait cycle of a representative PD patient, with the indication of the gait cycle phases. Figure 3B represents two examples of foot-switch signals acquired during atypical gait cycles (PFPS and PS) from a representative PD patient of the sample population.

The percentage of typical and atypical gait cycles compared to the total number of gait cycles, acquired during the walking task, was computed for each subject. For the typical gait cycles, we also analyzed the duration of each gait phase, expressed as the percentage of the Gait Cycle (%GC). For the atypical gait cycles, we analyzed their distribution over the walking path (i.e., percentage of atypical gait cycles acquired during the straight walk or the U-turns) and the most frequently observed atypical gait-cycle classes. The segmentation of the foot-switch signals into straight paths and U-turns was manually performed based on the video acquired during each experimental session. Since PD symptoms often begin on one side of the body, we separately analyzed the most affected side (first impaired lower limb), and the least affected one. In 9 out of 20 PD patients, the most affected side was the right one. In control subjects, we separately analyzed the dominant and non-dominant sides. The dominant side (i.e., the leg used to kick the ball) was the right one in 18 out of 20 control subjects.

Considering steady-state straight-line walking, in addition to gait speed, traditional spatio-temporal parameters were calculated (cadence and double support), as well as the duration of the 4 gait phases H, F, P, and S (in the typical gait cycles), for each lower limb.

Figure 4 shows a schematic description of the methodological steps implemented in this study.

### 2.4. Statistical Analysis

To determine if there is a statistically significant difference in walking speed, cadence, and double support between PD and control groups, the Lilliefors test (MATLAB^®^ function “*lillietest*”) was performed to test the normality of data distribution. Then, according to the results of the normality hypothesis test, the two-tailed Student’s *t*-test (in case of normal distribution) or the Wilcoxon signed-rank test (in case of non-normal distribution) was used, setting the significance level (α) at 0.05. Two-way analysis of variance (ANOVA) followed by post-hoc analysis with Tukey’s adjustment for multiple comparisons was used to test the differences in the gait analysis parameters between Group (PD and Control) and Side (more-affected and less-affected, or dominant and non-dominant). The effect size of the statistically significant differences was calculated through the Hedges’ *g* statistic, defined as in Equation (1), including the correction for small sample sizes (sample size ≤ 20) [43]:(1)$$g=\frac{M_{1}-M_{2}}{SD_{pooled}^{*}}=\frac{M_{1}-M_{2}}{\sqrt{\frac{\left(n_{1}-1\right)SD_{1}^{2}+\left(n_{2}-1\right)SD_{2}^{2}}{n_{1}+n_{2}-2}}}$$
where $M_{1}$ and $M_{2}$ are the averages of the two data distributions to be compared, $n_{1}$ and $n_{2}$ are the number of elements of each distribution, and $SD_{1}$ and $SD_{2}$ are their standard deviations.

The Hedge’s *g* statistic quantifies how much one sample distribution differs from another one in terms of number of standard deviations. According to the study by Hedge et al. [43], a *g* (absolute) value of approximately 0.2, 0.5, and 0.8 are considered as small, medium, and large effect size, respectively.

The statistical analysis was carried out using the Statistical and Machine Learning Toolbox of MATLAB^®^ release 2020b (The MathWorks Inc., Natick, MA, USA).

### 2.5. Correlation Analysis

We were also interested in studying the correlation between the percentage of atypical gait cycles extracted from the more-affected side of PD patients and the clinical scale UDPRS-III, widely used to evaluate parkinsonian motor impairment and disability. To obtain a better reproducibility of the study, we normalized the percentage of atypical gait cycles for each subject, with respect to individual walking speed, defining the following parameter:(2)$$Atyp=\frac{Percentage\;of\;atypical\;gait\;cycles}{Self-selected\;gait\;speed}$$

A correlation analysis between Atyp and UPDRS-III was performed, only for the PD patients, through linear regression. Starting from the regression line that best fits the sample data, Pearson’s correlation coefficient (*r*) was estimated. The 95% Confidence Interval (95% CI) for the estimated Pearson’s correlation coefficient was also obtained, using a bootstrap re-sampling technique. The bootstrap resampling technique is widely used to estimate descriptive statistics and confidence intervals when dealing with small sample populations (sample size ≤ 20). This procedure involves choosing random samples with replacement (i.e., each observation of the sample data is separately selected at random from the original dataset) from the sample data distribution to estimate the confidence interval for the parameters of interest. More specifically, the 95% CI of *r* was found by implementing the bootstrapping procedure in which the sample data were resampled 1000 times with replacement. The MATLAB^®^ function “*bootci*” was used to implement the bootstrapping procedure.

The correlation analysis was repeated twice, once considering the whole PD group (named Whole PD Population), and a second time considering a subset of PD patients, i.e., those showing an increased percentage of atypical cycles compared to the control group. This second group (named PD_with_Atyp) was obtained selecting PD patients with a normalized percentage of atypical cycles (Atyp) higher than the average level observed in the healthy population plus one standard deviation, defined as in (3):(3)$$Atyp>mean\left(Atyp_{Control}\right)+1\cdot std\left(Atyp_{Control}\right)$$
where $Atyp_{Control}$ is the normalized percentage of atypical cycles of control subjects (computed averaging over the dominant and non-dominant side).

## 3. Results

First, the comparison between PDs and controls is reported both for traditional spatio-temporal parameters and specifically considering atypical gait cycles and their characterization. Then, the correlation analysis (between the normalized percentage of atypical gait cycles and the clinical scale UDPRS-III) is reported for the PD cohort.

### 3.1. Gait Analysis: Classical Spatio-Temporal Parameters, Typical, and Atypical Gait Cycles

The straight-line walking speed, cadence, double support, as well as the total number of gait cycles analyzed, the percentage of typical gait cycles, the duration of each typical gait cycle phase, and the percentage of atypical gait cycles of the PD and control populations are reported in Table 1, with the indication of the statistically significant differences (*p* < 0.05).

On average, the walking speed, the cadence, and the double support of PD patients were not significantly different from that of control subjects (walking speed: *p* = 0.31; cadence: *p* = 0.53; double support: *p* = 0.13). Moreover, a similar number of gait cycles was analyzed in the two populations (*p* = 0.40).

Two-way ANOVA revealed a statistically significant decrease (*p* = 0.006, *g* = 0.64) in the percentage of typical gait cycles and a significant increase (*p* = 0.006, *g* = −0.64) in the percentage of atypical gait cycles of the PD group compared to the control group. Moreover, a slightly significant decrease (*p* = 0.036, *g* = 0.48) in the duration of the F-phase was found in PD patients compared to control subjects. No significant interaction effects were detected between Group and Side.

### 3.2. Atypical Gait Cycles during Straight Walking and U-Turning

The percentage of atypical gait cycles detected during straight walking and U-turning averaged over the PD and control populations are represented in Table 2, with the indication of the statistically significant differences (*p* < 0.05).

Two-way ANOVA showed statistically significant differences in the percentage of atypical gait cycles during straight walk (*p* = 0.007, *g* = −0.65) and U-turns (*p* < 0.0001, *g* = −1.26) between Group (PD and Control), while no significant differences were detected between Side (*p* = 0.12, and *p* = 0.36, respectively). No significant interaction effects were detected between Group and Side.

### 3.3. Characterization of Atypical Gait Cycles

The analysis of the most frequently observed foot-floor contact sequences revealed the existence of 10 different gait cycle classes (considering both the PD and control populations): The typical class of HFPS gait cycles and 9 atypical classes of gait cycles (i.e., gait cycles showing foot-floor contact sequences different from HFPS). The most common atypical cycles were:PFPS and PS: gait cycles characterized by forefoot initial contact;FPS: gait cycles characterized by flat-foot initial contact;HFHFPS: gait cycles characterized by an unstable heel contact (although heel strike is present).

Figure 5 shows the boxplots of the percentage of atypical cycles relative to the 9 most frequently observed atypical foot-floor contact sequences, in PDs and controls.

Two-way ANOVA showed statistically significant differences in the percentage of PFPS (*p* = 0.006, *g* = −0.63) and PS (*p* = 0.013, *g* = −0.56) gait cycles between Group (PD and Control), while no significant differences were detected between Side (*p* = 0.38 and *p* = 0.18, respectively). No significant interaction effects were detected between Group and Side.

In particular, an average percentage of PFPS gait cycles of 6.7% ± 7.3% and 5.8% ± 1.0% was computed for the more-affected and less-affected side of the PD population, respectively, while an average value of 3.7% ± 0.8% and 2.7% ± 2.9% was obtained for the dominant and non-dominant side of the control group, respectively. Considering the PS gait cycles, an average percentage of 6.1% ± 9.5% and 2.0% ± 0.9% was obtained for the more-affected and less-affected side of the PD population, respectively, while an average value of 0.4% ± 0.3% and 1.1% ± 3.9% was computed for the dominant and non-dominant side of the control group, respectively.

### 3.4. Correlation Analysis Considering UPDRS-III

In this section, we present the results of the correlation analysis obtained considering: (i) the whole PD group, and (ii) the subgroup PD_with_Atyp. This latest is composed of patients showing a value of Atyp higher than 17.5% (“control level” defined by Equation (3)). Eight out of twenty PD patients satisfied this condition.

Figure 6 reports the correlation analysis between Atyp (more-affected side) and UPDRS-III, including the bootstrap resampling histograms for the whole PD population (Figure 6A) and the PD_with_Atyp subgroup (Figure 6B).

Considering the whole PD group, a moderate correlation (*r* = 0.60, *p* = 0.005) was found between Atyp (more-affected side) and UPDRS-III (95% CI: [−0.07, 0.88]). Considering the subgroup PD_with_Atyp, a strong correlation was found (*r* = 0.91, *p* = 0.002) between Atyp (more-affected side) and UPDRS-III (95% CI: [0.59, 0.98]).

## 4. Discussion

Gait disturbance is a key component of motor impairment in parkinsonian patients. Therefore, it is not surprising that a large body of experimental work was aimed at describing and monitoring locomotion abnormalities in PD, most often through automatic motion analysis techniques [10,14,24,26,27,28,44,45,46,47,48].

In this work, we proposed and validated the percentage of atypical gait cycles (defined as those gait cycles showing a foot-floor contact sequence different from HFPS) as a motor biomarker that is useful for an objective assessment of the gait performance of PD patients, specifically focusing on the foot-floor contact quality, which is related to distal motor control.

It is worth highlighting that when the foot–floor contact sequence is studied, an inaccurate evaluation of the gait events can be particularly detrimental in the clinical setting applications. Furthermore, the identification of the gait events in neurological disorders can be more complex than in healthy subjects with intact locomotor function, specifically for the way in which the foot first touches the floor at the beginning of each gait cycle. As an example, the “normal” rearfoot strike (i.e., the heel touches the ground first) may be partially or totally missing in some patients, due to an impaired control of the ankle joint. Depending on the severity of the motor dysfunction, a specific portion of the patients’ strides (or the wholeness of the strides) can display an alternative (pathological) sequence of foot-floor contact. Indeed, these abnormal strides can initiate with a forefoot strike, usually complicating the analysis. However, systems specifically designed for clinical gait analysis can overcome this challenge [29].

The signals acquired in this study were collected through a portable, lightweight, low-cost system for clinical gait analysis exploiting foot-switches [49]. Many different systems based on body-worn sensors are employed for diagnostic and monitoring applications in parkinsonian patients [10,50,51]. The large majority of instrumented gait studies on PDs carries out measurements, either by means of systems based upon IMU sensors [14,52,53] and/or foot Force Sensing Resistors (FSR) of various kinds [28,54]. The wide variety of sensors used for detecting gait events, and timing the gait cycle phases, fall into two broad categories: Sensors that allow for a direct measure of the gait events (foot-switches and FSR) and those that indirectly reconstruct the timing of these events (IMUs) [21]. Indeed IMUs, i.e., accelerometers, gyroscopes, and magnetometers, necessitate customized signal processing to gain information on gait events and, hence, model reconstruction errors may be intrinsically higher. Not only they often require cohort-specific algorithms to tackle the challenges of studying gait events in neurological populations, but also different processing strategies are required to cope with straight path and turnings. For this reason, our strategy was to directly record the foot–floor contact sequence of PDs (and age-matched controls) through foot-switches.

Foot-switches are easily mounted directly under the barefoot sole (through bi-adhesive tape), without the need for dedicated shoes or sensing insoles. The time to prepare the subject is usually less than five minutes, which is fully compatible with clinical requirements. The selection of patient-specific anatomical landmarks under the foot (heel, first and fifth metatarsal heads, these latest simply identified through bone palpation) avoids sensorized shoes or shoe-insoles from being worn differently by specific patients and/or that sensors move with respect to the foot during gait. Moreover, foot deformities are not rare in parkinsonian patients.

Another aspect to be considered is that a single foot-switch provides a signal for either foot contact (on) or lack of contact (off). This binary characteristic directly produces a “sharp” timing of the gait phases, useful in gait event detection. Other techniques, e.g., those relying on foot plantar pressure measurements [55], are based on many small sensors that allow for detailed monitoring of the foot–floor pressure. However, they do not directly provide sharp indications on the sub-phases of stance. Typically, some pre-processing of the insole pressure signal is required [18].

### 4.1. Gait Analysis: Classical Spatio-Temporal Parameters, Typical and Atypical Gait Cycles

On average, we analyzed more than 500 gait cycles for each subject, considering both lower limbs. In the PD population, which we examined during best-ON conditions, the straight-line walking speed was not different from that of controls (PD speed: 1.01 ± 0.25 m/s; controls’ speed: 1.08 ± 0.17 m/s; *p* = 0.31). The overall number of gait cycles recorded during the 5-min walking task was also similar (PDs: 269 ± 47 and 278 ± 38 gait cycles were recorded on the most and less affected side, respectively; Controls: 267 ± 16 and 268 ± 19 gait cycles were recorded on the dominant, and non-dominant side, respectively; *p* = 0.40). The fact that the straight-line speed and the total number of recorded gait cycles were similar between PD and control populations allows us for directly comparing their gait parameters, since no bias arising from gait speed or mismatch in the number of strides collected is demonstrated. On the other hand, this provides indirect evidence that, at least in their best-ON condition, the PD individuals belonging to the examined population showed only very mild gait dysfunctions, which is not detectable considering straight-line speed or the overall number of gait cycles traveled in five minutes. In addition, considering traditional spatio-temporal parameters, we also found that cadence and double support of the examined parkinsonian patients were not altered, during steady-state straight-line walking (PD cadence: 55.7 ± 5.9 cycles/min, controls’ cadence: 54.6 ± 3.3 cycles/min, *p* = 0.53; PD double support: 11.9 ± 5.5%GC, controls’ double support: 14.2 ± 3.9%GC, *p* = 0.13). This distinguishes our study from those of others, also considering mild PD dysfunctions. As an example, the work of Ref. [47] reports a reduction of the average walking speed and cadence of early-stage PD patients (seven individuals), and discussed their tendency to bradykinetic gait along a straight trajectory of about 6 m. That work also found a similar double-support timing between PD and controls, in accordance with the present study. This work highlighted the importance of studying atypical gait cycles, in addition to typical ones.

Considering typical gait cycles (HFPS, characterized by normal heel strike), we found only a small decrease of the flat-foot contact (F) duration in PD patients compared to controls, at the limit of the statistical significance (*p* = 0.04). The F-phase duration decreased only by 4%GC (value obtained by averaging the two lower limbs). Notice that it was possible to reveal this subtle difference because we considered, separately, the sub-phases of stance (heel contact, flat-foot contact, and push-off), instead of estimating the whole stance-phase duration (H+F+P). While traditional gait analysis typically provides measurements of only two main phases (stance and swing), the analysis of the 4-level basography allows for timings four gait phases, for each lower limb. Separating stance into its sub-phases often provides more sensitive outcome measures, as it was already suggested in previous studies related to other neurological populations, e.g., hemiplegic children after cerebral palsy [32,56,57], and patients affected by mild ataxia [30]. Furthermore, it is advisable to adopt gait partitioning methods that do not require pathology-specific templates, or the a-priori definition of the number of gait phases. Even if the most common approach is based on the four-phase model, models with a different number of gait phases are also used [58,59,60].

Considering atypical cycles, the differences between PDs and controls become definitely more evident. We found a consistent increase (+42%, *p* = 0.006) in the total percentage of atypical cycle of parkinsonian patients with respect to controls (PD more affected side: 25.4% ± 21.5%, PD less affected side: 15.5% ± 10.1%; controls’ dominant side: 8.1% ± 5.6%, controls’ non-dominant side: 8.8% ± 4.9%).

In the present work, we focused on foot-floor contact analysis. Future studies can advantageously consider the combined investigation of foot-floor contact sequences and ankle kinematics [32,61].

### 4.2. Atypical Gait Cycles during Straight Walking and U-Turning

This study suggests that the percentage of the atypical gait cycle is a sensitive parameter, able to grasp subtle deviation from normality, not detectable by traditional spatio-temporal parameters used in gait analysis.

We found a significant increase in the percentage of atypical cycles in PD patients during both the straight path and U-turns. Averaging both sides, the PD patients showed overall 8.9% atypical gait cycles during straight walk (PD more-affected side: 13.4% ± 18.3%, PD less-affected side: 4.4% ± 7.8%) and 11.9% atypical gait cycles during U-turns (PD more-affected side: 13.1% ± 6.1%, PD less-affected side: 10.7% ± 4.4%). Averaging the dominant and non-dominant side, the control population showed overall 2.3% atypical gait cycles during straight walk (controls’ dominant side: 2.0% ± 2.5%, controls’ non-dominant side: 2.5% ± 3.4%) and 6.2% atypical gait cycles during U-turns (controls’ dominant side: 6.1% ± 4.1%, controls’ non-dominant side: 6.3% ± 3.0%).

Gait cycles initiating with a forefoot strike (PFPS and PS), instead of a heel strike, are among the most frequently observed atypical cycles in the PD group, and their percentage is significantly increased compared to controls (PFPS in PDs: 6.7% ± 7.3% and 5.8% ± 1.0% for the more-affected and less-affected side, respectively, PFPS in controls: 3.7% ± 0.8% and 2.7% ± 2.9% for the dominant and non-dominant side, respectively, *p* = 0.006; PS in PDs: 6.1% ± 9.5% and 2.0% ± 0.9% for the more-affected and less-affected side, respectively, PS in controls: 0.4% ± 0.3% and 1.1% ± 3.9% for the dominant and non-dominant side, respectively, *p* = 0.013). The authors hypothesize that the presence of forefoot-strike cycles, and in particular, PS cycles (where the heel never touches the ground during the entire stride), can be related to an increase in fall risk. However, further research is required to provide evidence in this regard.

### 4.3. Correlation Analysis Considering UPDRS-III

Furthermore, this study confirmed the validity of the use of the normalized the percentage of atypical gait cycles in the discrimination of PD patients with different degrees of motor impairment. We found that when atypical gait cycles are present in PD patients in an increased amount compared to a control population, their normalized percentage is strongly correlated with UPDRS-III (*r* = 0.91, 95% CI: [0.59, 0.98], *p* = 0.002).

## 5. Conclusions

The percentage of atypical cycles proved to be a valid biomarker to quantify subtle gait dysfunctions in parkinsonian patients. Besides the description and characterization of the atypical gait cycles in PD, control subjects and their comparison, the novelty of this work resides in the large number of gait cycles analyzed along a continuous motor task lasting five minutes, which includes both straight-line walking and U-turnings. This kind of protocol enabled the detection of very mild gait abnormalities in the distal motor control of PD patients, not perceptible by traditional spatio-temporal parameters used in gait analysis. Through the segmentation of foot-switch signals (directly providing information on gait events), it was possible to measure the occurrence of different foot-floor contact sequences. Furthermore, we demonstrated that, when there is an abnormal percentage of atypical gait cycles in a parkinsonian subject, this objective parameter is strongly correlated with UPDRS-III, which is commonly used by neurologists for (subjectively) assessing the PD motor dysfunctions. This documents the applicability of the proposed methodology in the clinical management of PD patients.

## Figures and Tables

**Figure 1 sensors-21-05079-f001:**
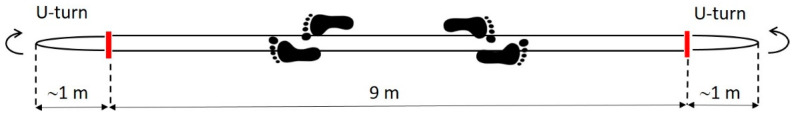
Schematic representation of the walking path. Subjects walked back and forth, without interruptions, along a straight path of 9 m, for approximately 5 min. U-turns were included in the analysis.

**Figure 2 sensors-21-05079-f002:**
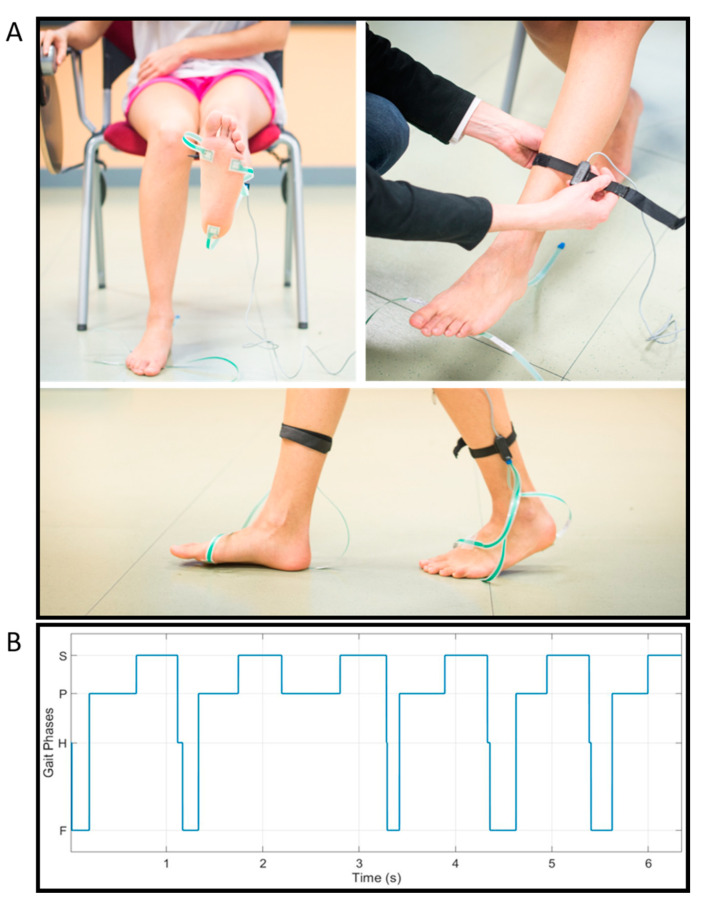
Panel (**A**) represents the acquisition system. Foot-switches were placed under the heel, the first, and the fifth metatarsal heads of each foot to detect the foot-floor contact events. Panel (**B**) shows an example of foot-switch signal acquired from the more-affected side of a representative PD patient of the sample population with the indication of the four gait phases (H: Heel Contact, F: Flat-Foot Contact, P: Push-Off, and S: Swing).

**Figure 3 sensors-21-05079-f003:**
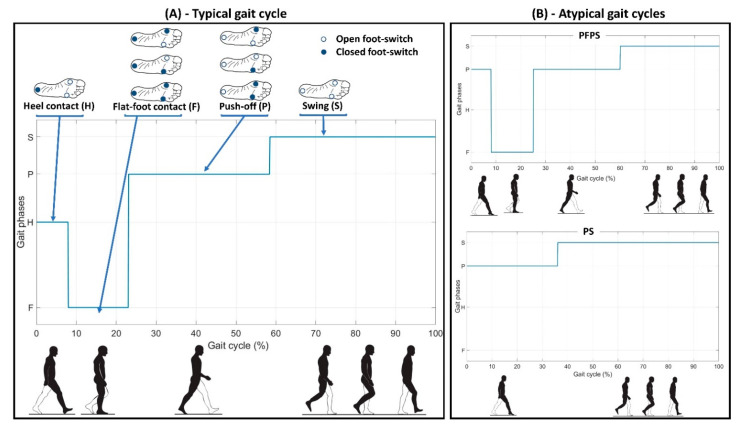
Example of 4-level foot-switch signals and gait cycle phases (H: Heel Contact, F: Flat-Foot Contact, P: Push-Off, and S: Swing) acquired from the right foot (more affected side) of a representative PD patient. Panel (**A**) represents an example of a typical gait cycle, while panel (**B**) shows two examples of atypical gait cycles (PFPS and PS).

**Figure 4 sensors-21-05079-f004:**
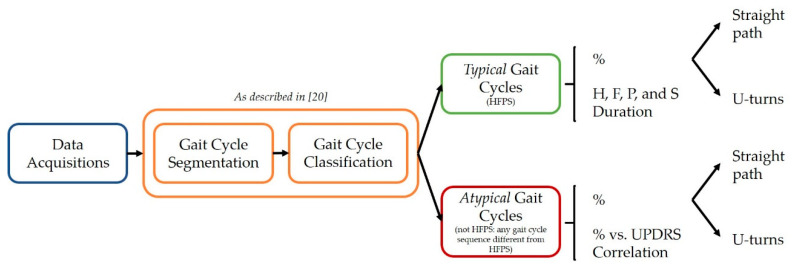
Schematic representation of the methodological steps implemented in this study from the data acquisition step to the assessment of the results.

**Figure 5 sensors-21-05079-f005:**
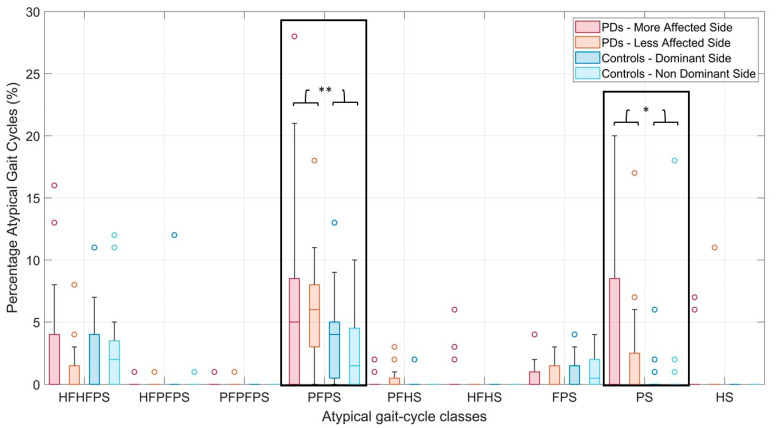
Boxplots of the percentage of atypical gait cycles (i.e., any gait cycle showing a foot-floor contact sequence different from HFPS) relative to the most frequently observed atypical gait-cycle classes in PD and control populations. Outliers are indicated by circles. Single and double asterisks represent statistically significant differences with *p*-values lower than 0.05 and 0.01, respectively.

**Figure 6 sensors-21-05079-f006:**
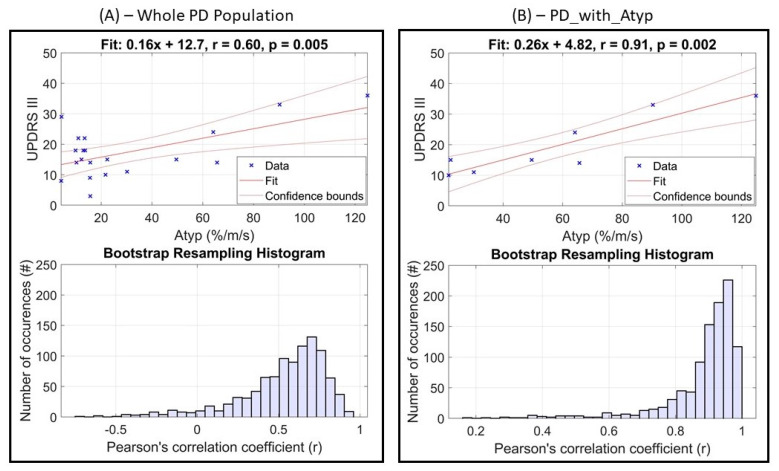
Correlation analysis between the normalized percentage of atypical gait cycles (for the more-affected side of PD patients) and UPDRS-III computed for: (**A**) the whole PD group, and (**B**) the PD_with_Atyp subgroup.

**Table 1 sensors-21-05079-t001:** Gait analysis parameters averaged over Parkinson’s Disease (PD) patients and control subjects.

	PD Patients		Control Subjects		Wilcoxon Test (*p*-Value)	
Walking speed (m/s)	1.01 ± 0.25		1.08 ± 0.17		0.31	
Cadence (cycles/min)	55.7 ± 5.9		54.6 ± 3.3		0.53	
Double support (%GC)	11.9 ± 5.5		14.2 ± 3.9		0.13	
	**PD patients**		**Control subjects**		**2-way ANOVA** **(*p*-value)**	
	**More-affected side**	**Less-affected side**	**Dominant side**	**Non-dominant side**	**Group**	**Side**
Total number of gait cycles	269 ± 47	278 ± 38	267 ± 16	268 ± 19	0.40	0.69
***Typical*** **gait cycles**						
Percentage of HFPS (%)	**74.6 ± 21.5**	**84.5 ± 10.1**	**91.9 ± 5.6**	**91.2 ± 4.9**	**0.006**	0.13
H-phase duration (%GC)	9.6 ± 6.8	9.1 ± 6.0	7.1 ± 1.8	9.9 ± 5.1	0.53	0.34
F-phase duration (%GC)	**20.3 ± 10.4**	**22.2 ± 7.9**	**25.0 ± 4.8**	**24.9 ± 6.4**	**0.036**	0.60
P-phase duration (%GC)	25.5 ± 8.2	24.7 ± 5.9	24.4 ± 5.0	22.8 ± 3.9	0.28	0.38
S-phase duration (%GC)	44.6 ± 4.3	44.0 ± 5.1	43.4 ± 2.4	42.3 ± 2.4	0.09	0.31
***Atypical*** **gait cycles**						
Percentage of *atypical* GC (%)	**25.4 ± 21.5**	**15.5 ± 10.1**	**8.1 ± 5.6**	**8.8 ± 4.9**	**0.006**	0.13

*Atypical* gait cycles are defined as the gait cycles showing a foot-floor contact sequence different from HFPS. Values of parameters are reported as mean ± standard deviation over the sample populations. Statistically significant differences (*p* < 0.05) between Group or Side are represented in bold. Group: PD patients vs. Control subjects. Side: More-affected vs. less-affected side of PD patients; Dominant vs. Non-dominant side of Control subjects. %GC: Percentage of Gait Cycle.

**Table 2 sensors-21-05079-t002:** Percentage of atypical gait cycles in straight path and U-turns in PD patients and control subjects.

Percentage of *atypical* GC (%)	PD Patients		Control Subjects		2-way ANOVA (*p*-Value)	
	More-Affected Side	Less-Affected Side	Dominant Side	Non-Dominant Side	Group	Side
Straight walking	**12.3 ± 18.3**	**4.8 ± 7.8**	**2.0 ± 2.5**	**2.5 ± 3.4**	**0.007**	0.12
U-turning	**13.1 ± 6.1**	**10.7 ± 4.4**	**6.1 ± 4.1**	**6.3 ± 3.0**	**<0.0001**	0.36

*Atypical* gait cycles are defined as the gait cycles showing a foot-floor contact sequence different from HFPS. Values of parameters are reported as mean ± standard deviation over the sample populations. Statistically significant differences (*p* < 0.05) between Group or Side are represented in bold. Group: PD patients vs. Control subjects. Side: More-affected vs. less-affected side of PD patients; Dominant vs. Non-dominant side of Control subjects. %GC: Percentage of Gait Cycle.

## Data Availability

Data and materials presented in this study are available on reasonable request from the corresponding author.

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
