# Peer review of "Atypical Gait Cycles in Parkinson’s Disease"

_sensors, 2021, doi:10.3390/s21155079_

Round 1
Reviewer 1 Report
The paper proposes a novel objective biomarker for PD gait, the increase in atypical gait cycles in PD patients compared to healthy controls. The authors performed gait analysis using foot-switches sensors fixed directly on barefoot sole at precise anatomical landmark, connected to a multichannel acquisition system placed on the subject’s leg. They analyzed gait cycle of PD patients in best ON conditions both on a stride path and in U-turns. They defined typical gait cycle as the gait characterized by the physiological sequence of Heel Contact, Flat Foot Contact, Push Off and Swing, while any other sequence defines an atypical gait cycle. The authors demonstrated that the increased percentage of atypical gait cycle is a sensitive parameter that distinguish PD patients from controls. This biomarker correlates with the degree of motor impairment, measured by UPDRS-III. The topic is of great interest and the manuscript provide a novelty in the field. Methods and statistical analysis are adequate, the results are clearly exposed, the discussion and conclusion are relevant. In my opinion the present manuscript is suitable for publication in Sensors without any further revision.
Reviewer 2 Report
This article focuses on the characterization of atypical gait cycles in Parkinson's disease patients compared to healthy subjects. The topic covered in the article is of interest. However, I recommend that the authors make the following changes to improve the quality of the manuscript:
The introduction correctly describes the background of the investigation.
The authors make a very simple description of the gait phases: Contact with the heel (H), Contact with flat foot (F), Push off (P) and Swing (H). In this way, we consider that the analysis of the gait is not very specific and that information about the different phases is lost. In this sense, authors should consider Rancho Los Amigos (RLA) terminology.
In Material and Methods, the authors should quantify the cognitive status of the participants. It is a fundamental variable that influences gait, mainly in the cognitive analysis of turns.
I believe that the analysis of the gait phases should be adapted to the aforementioned terminology. The authors can analyze only 4 phases but it is important to adapt the terminology: loading response, middle support, terminal support, etc.
Regarding how to detect the phases, the authors should consider other criteria. For example, force recorded in Newtons performed on each sensor. Is it possible to measure it?
It is not clear in the methodology what an atypical gait cycle is. Are the authors based on the sequence of phases? Or do they attend to spatio-temporal or kinematic parameters?
In results, it is important to re-describe the acronyms of atypical gait cycles, both in the text and in the figure caption. From my point of view, the description of the atypical patterns is not clear. In the clinical setting, it is important to define the pattern that the foot performs.
In discussion, it remains unclear to me what an atypical gait cycle is.
In general terms, patients can modify their contact for anthropometric reasons (flat feet, cavus, pronated, etc.) or they can make a correct contact without having adequate kinematics in the rest of the joints. In this sense, I consider that it would be interesting to correlate the findings recorded in the foot with the articular kinematics, through an observational scale or a low-cost system such as Kinovea.
Reviewer 3 Report
- Lines (163-164): More details of your anti-causal anti-bounce filter is highly welcomed.
- At the table 1 you have written that the percentage of HFPS (typical gait cycles) is no more than only 92 % for healthy subjects. Please comment this results of text. Is this the limitation of your method?
- What about the gait cycles: HFHFPS which have been observed for control group (fig. 4). How did the healthy subjects during gait in phase Flat Foot reach phase Heel Strike? Do you know any other papers which report this phenomenon? Did you validate it with any of capture motion system? What was the second H-phase average duration (as % of gait cycle)?
Minor:
- (Line 211) It could be useful to add the formula for calculation of the pooled weighted standard deviation.
- It could be useful to add a graph to present all the steps which have been done in preprocessing data: from acquiring data to obtained results
- (Caption of the table2.) ‘percentage’ is more suitable word than ‘distribution’ in this case.
- Is “Percentage of atypical GC(%)” equal 100% - “Percentage of HFPS (%)” ? Please verify the values for PD patients (table1 vs table 2)
- t should be underlined that dominant/non-dominant side is for healthy subjects and more-affected/ less-affected for the PD ones. Please add the proper information at the table 1 (eg. like at the table 2) or in the text.
Round 2
Reviewer 2 Report
Dear authors, I have carefully read the changes that you have made . I believe that the manuscript has improved and can be accepted.
Best regards
Author Response
Answers to Reviewer #2
Manuscript ID: sensors-1258852
Manuscript title: Atypical gait cycles in Parkinson’s disease
GENERAL COMMENTS:
Dear authors,
I have carefully read the changes that you have made. I believe that the manuscript has improved and can be accepted.
Best regards
We thank Reviewer #2 for the time spent reading the amended manuscript and the point-by-point response. We are glad that Reviewer #2 found the manuscript quality improved and suitable for publication in Sensors.

Reviewer 3 Report
Thank you for your improvements which have been made in the paper as well as the answers in cover letter. However I still have doubts connected with HFHFPS gait cycles which have been observed in gait of healthy subject. According to my suggestion you have referred to other papers but the coauthor of those paper is Mrs. Agostini who is the coauthor of this paper too. So I would like to ask you one more time to refer to papers of other authors who observed HFHFPS gait cycle for healthy subjects.
Author Response
Answers to Reviewer #3
Manuscript ID: sensors-1258852
Manuscript title: Atypical gait cycles in Parkinson’s disease
GENERAL COMMENT:
Thank you for your improvements which have been made in the paper as well as the answers in cover letter.
We thank Reviewer #3 for the time spent reading the amended manuscript and the point-by-point response. We are glad that Reviewer #3 found the manuscript quality improved compared to the original version.
SPECIFIC COMMENT:
- However I still have doubts connected with HFHFPS gait cycles which have been observed in gait of healthy subject. According to my suggestion you have referred to other papers but the coauthor of those paper is Mrs. Agostini who is the coauthor of this paper too. So I would like to ask you one more time to refer to papers of other authors who observed HFHFPS gait cycle for healthy subjects.
As suggested, we added further literature to consider other papers introducing 6 gait phases instead of the traditional 4 gait phases (page 2, line 52 and page 14, lines 442 - 446):
[20] J. Taborri, E. Palermo, S. Rossi, and P. Cappa, “Gait partitioning methods: A systematic review,” Sensors (Switzerland), vol. 16, no. 1, pp. 1–20, 2016, doi: 10.3390/s16010066.
[56] J. Taborri, E. Scalona, E. Palermo, S. Rossi, and P. Cappa, “Validation of Inter-Subject Training for Hidden Markov Models Applied to Gait Phase Detection in Children with Cerebral Palsy,” Sensors 2015, Vol. 15, Pages 24514-24529, vol. 15, no. 9, pp. 24514–24529, Sep. 2015, doi: 10.3390/S150924514.
[57] J. Taborri, E. Scalona, S. Rossi, E. Palermo, F. Patane, and P. Cappa, “Real-time gait detection based on Hidden Markov Model: Is it possible to avoid training procedure?,” 2015 IEEE Int. Symp. Med. Meas. Appl. MeMeA 2015 - Proc., pp. 141–145, Jun. 2015, doi: 10.1109/MEMEA.2015.7145188.
[58] J. Taborri, S. Rossi, E. Palermo, F. Patanè, and P. Cappa, “A Novel HMM Distributed Classifier for the Detection of Gait Phases by Means of a Wearable Inertial Sensor Network,” Sensors 2014, Vol. 14, Pages 16212-16234, vol. 14, no. 9, pp. 16212–16234, Sep. 2014, doi: 10.3390/S140916212.
[59] A. C. Villa-Parra, D. Delisle-Rodriguez, J. S. Lima, A. Frizera-Neto, and T. Bastos, “Knee impedance modulation to control an active orthosis using insole sensors,” Sensors (Switzerland), vol. 17, no. 12, 2017, doi: 10.3390/s17122751.
[60] A. C. Villa-Parra, J. Lima, D. Delisle-Rodriguez, L. Vargas-Valencia, A. Frizera-Neto, and T. Bastos, “Assessment of an assistive control approach applied in an active knee orthosis plus walker for post-stroke gait rehabilitation,” Sensors (Switzerland), vol. 20, no. 9, pp. 1–15, 2020, doi: 10.3390/s20092452.
To the best of our knowledge, there are no works in literature specifically focusing on HFPSPS cycles, both on healthy or pathological subjects.
In healthy subjects, the number of HFHFPS cycles is very small: they are less than 1% of the total number of collected gait cycles. More specifically, we observed 23 HFHFPS cycles out of 5347 total gait cycles (0.42%) on the dominant side, and 33 HFHFPS cycles out of 5359 total gait cycles (0.61%) on the non-dominant side of healthy subjects. Almost all of these gait cycles were observed during U-turns/transitions from U-turns to straight path (and viceversa). A healthy-subject's HFHFPS cycle is provided in the figure below (Figure 1) as an example.
(see the attached document "Answers to Reviewer3 - R2.docx")
Figure 1 | Representative HFHFPS gait cycle acquired from a healthy subject of the sample population during U-turn/Straight path transition. The first and the third channels represent, respectively, the foot-switch signals acquired from the non-dominant (left) and dominant side (right), while the second and the fourth channels represent the knee joint flexion/extension angles of the non-dominant and dominant side, respectively. Blue vertical lines represent the beginning and the end of the U-turn, while the yellow rectangle highlights the HFHFPS gait cycle.
This cycle is most probably related to the subject pivoting on the right foot to negotiate the U-turn (as it can be noticed from the right knee joint kinematics). There was no statistical difference between PD and controls for this “rare” foot pattern. Due to the very limited presence of HFHFPS cycles during the 5-minutes walk, we did not specifically focus on this kind of gait cycles in our analysis.
